# INX-18 and INX-19 play distinct roles in electrical synapses that modulate aversive behavior in *Caenorhabditis elegans*

**Lisa Voelker**[1,2], **Bishal Upadhyaya**[1], **Denise M. Ferkey**[3], **Sarah Woldemariam**[4], **Noelle D. L'Etoile**[4], **Ithai Rabinowitch**[1,5], **Jihong Bai**[1,2] *

**1** Basic Sciences Division, Fred Hutchinson Cancer Research Center, Seattle, WA, United States of America, **2** Molecular and Cellular Biology Program, University of Washington, Seattle, WA, United States of America, **3** Department of Biological Sciences, University at Buffalo, The State University of New York, Buffalo, NY, United States of America, **4** Department of Cell and Tissue Biology, University of California, San Francisco, CA, United States of America, **5** Department of Medical Neurobiology, Faculty of Medicine Hebrew, University of Jerusalem, Jerusalem, Israel

* jbai@fredhutch.org

**Data Availability Statement:** All relevant data are within the manuscript and its Supporting Information files.

## Abstract

In order to respond to changing environments and fluctuations in internal states, animals adjust their behavior through diverse neuromodulatory mechanisms. In this study we show that electrical synapses between the ASH primary quinine-detecting sensory neurons and the neighboring ASK neurons are required for modulating the aversive response to the bitter tastant quinine in *C. elegans*. Mutant worms that lack the electrical synapse proteins INX-18 and INX-19 become hypersensitive to dilute quinine. Cell-specific rescue experiments indicate that *inx-18* operates in ASK while *inx-19* is required in both ASK and ASH for proper quinine sensitivity. Imaging analyses find that INX-19 in ASK and ASH localizes to the same regions in the nerve ring, suggesting that both sides of ASK-ASH electrical synapses contain INX-19. While *inx-18* and *inx-19* mutant animals have a similar behavioral phenotype, several lines of evidence suggest the proteins encoded by these genes play different roles in modulating the aversive quinine response. First, INX-18 and INX-19 localize to different regions of the nerve ring, indicating that they are not present in the same synapses. Second, removing *inx-18* disrupts the distribution of INX-19, while removing *inx-19* does not alter INX-18 localization. Finally, by using a fluorescent cGMP reporter, we find that INX-18 and INX-19 have distinct roles in establishing cGMP levels in ASK and ASH. Together, these results demonstrate that electrical synapses containing INX-18 and INX-19 facilitate modulation of ASH nociceptive signaling. Our findings support the idea that a network of electrical synapses mediates cGMP exchange between neurons, enabling modulation of sensory responses and behavior.

## Author summary

Animals are constantly adjusting their behavior to respond to changes in the environment or to their internal state. This behavior modulation is achieved by altering the activity of

**Funding:** This research was supported by
R21DC016158 (JB) (National Institutes of Health,
https://grants.nih.gov/grants/funding/r21.htm),
R01DC015758 (DMF), R01DC005991 (NDL),
3R01DC005991-11A1S1 (SW) (National Institutes
of Health, https://grants.nih.gov/grants/funding/
r01.htm) and PHS NRSA T32GM007270 (LV)
(National Institutes of Health, https://
researchtraining.nih.gov/programs/training-grants/
t32). The funders had no role in study design, data
collection and analysis, decision to publish, or
preparation of the manuscript.

**Competing interests:** The authors have declared
that no competing interests exist.

neurons and circuits through a variety of neuroplasticity mechanisms. Chemical synapses are known to impact neuroplasticity in several different ways, but the diversity of mechanisms by which electrical synapses contribute is still being investigated. Electrical synapses are specialized sites of connection between neurons where ions and small signaling molecules can pass directly from one cell to the next. By passing small molecules through electrical synapses, neurons may be able to modify the activity of their neighbors. In this study we identify two genes that contribute to electrical synapses between two sensory neurons in *C. elegans*. We show that these electrical synapses are crucial for proper modulation of sensory responses, as without them animals are overly responsive to an aversive stimulus. In addition to pinpointing their sites of action, we present evidence that they may be contributing to neuromodulation by facilitating passage of the small molecule cGMP between neurons. Our work provides evidence for a role of electrical synapses in regulating animal behavior.

## Introduction

A defining feature of animal behavior is its plasticity. Animals adapt their behavior in order to respond to environmental challenges and physiological changes. Such behavioral plasticity is essential for animal survival and is achieved by changing the activity of neurons and circuits in a variety of ways. One way is through neuromodulation, whereby diffusible signals such as neuropeptides, dopamine, and serotonin are used to tune brain activity in broad regions [1–3]. By contrast, neuronal activity can be altered locally by changing the strength of individual synapses [4, 5]. In order to understand dynamic brain function, it is crucial to uncover mechanisms that drive neuroplasticity at various levels.

Electrical synapses (also known as gap junctions) are composed of membrane channels that join the cytoplasm of two cells [6]. They are found throughout vertebrate and invertebrate nervous systems [6–9] where they pass both electrical and chemical signals between connected cells [10]. Electrical synapses have been primarily studied for their ability to synchronize electrical activity between pairs or groups of neurons [11–13], but can also pass small molecules such as calcium [14, 15], cAMP [16–19], cGMP [17, 20], $IP_3$ [15, 21], and even small miRNA [22, 23]. Interestingly, while electrical synapses share similar function and protein topology in vertebrates and invertebrates [24], genes encoding electrical synapse components are evolutionarily unrelated [6, 10]. As a result, electrical synapses in vertebrates are composed of connexins, while those in invertebrates are composed of innexins (INXs). The separate evolution of electrical synapses suggests the functional necessity of these channels, although their role in neural plasticity and brain function is not fully understood.

Recently, it was discovered that innexin networks play a crucial role in cGMP-dependent sensory modulation in *Caenorhabditis elegans* [25]. Krzyzanowski and colleagues found that cGMP functions within the sensory neuron ASH to dampen nociceptive sensitivity but is produced in neighboring neurons [26]. They further showed that cGMP-mediated dampening of ASH nociceptive sensitivity requires an innexin-based network [25]. These findings uncover a new strategy of network regulation that may contribute to the modulation of neural activity. ASH is the primary nociceptive neuron pair in *C. elegans* and responds with increased calcium levels to diverse aversive stimuli including hyperosmolarity, nose touch, heavy metals such as copper, volatile repellents such as octanol and alkaloids such as quinine [27–33]. ASH controls movement away from noxious stimuli through synapses on the forward and backward command interneurons [34, 35]. Nociception in ASH is extensively modulated, and reactivity to

aversive stimuli such as quinine is regulated by the presence of food and the satiety state of the worm [25, 36–40]. Notably, ASH forms electrical synapses with multiple other sensory neurons and a few interneurons [41, 42], suggesting electrical synapses may be crucial in modulating its activity.

We investigated the impact of electrical synapses between ASH and its neighbor ASK on behavioral sensitivity to the bitter tastant quinine. ASK forms multiple electrical synapses with ASH [42] and expresses several innexins [8, 43, 44], making it a candidate for directly modifying ASH activity. Results of this study show that the electrical synapse proteins INX-18 and INX-19 function within ASK and ASH to allow for modulation of the quinine avoidance response. Through imaging, we found that INX-18 and INX-19 localize to known sites of electrical synapses. Our data further suggest that INX-19 plays a principle role in diffusion of cGMP from ASK to ASH. Our study identifies a direct connection between two sensory neurons that modulates neuronal activity and thus regulates behavior in *C. elegans*.

## Results

### Innexin-18 and innexin-19 are required for modulation of the quinine response

A recent study suggests that a network of electrical synapses is involved in modulation of the quinine response [25], however the exact composition of those electrical synapses has not been determined. ASH is a multimodal nociceptive neuron that responds to quinine and forms direct electrical synaptic connections with the sensory neuron ASK [41, 42], which is also involved in quinine sensation [32]. To explore whether the electrical synapses between ASK and ASH play a role in modulating quinine sensitivity, we investigated the innexins INX-18 and INX-19 that are expressed in these two sensory neurons [8, 43, 44]. While INX-4 is also expressed in ASH, we did not include it in our analyses as it has already been explored in a previous study [25].

To determine whether INX-18 and/or INX-19 play a role in modulating the behavioral response to quinine, we assayed *inx-18(ok2454)*, *inx-19(ky634)* and *inx-19(tm1896)* mutant animals (Fig 1A and 1B) for quinine sensitivity. We placed drops of quinine solution in front of freely crawling worms and recorded their responses as "responding" if they reverse or "non-responding" if they continue forward [32, 45]. We found that these mutant animals were hypersensitive to 1 mM quinine in the quinine drop test (Fig 1C). As a negative control, we examined the response of mutant animals to M13 buffer. Both *inx-18(ok2454)* and *inx-19 (tm1896)* animals responded to M13 buffer at similar levels to wild-type (N2) animals, *inx-19 (ky634)* animals, however, were slightly more responsive than wild-type animals (S1A Fig). This may be because this strain has mildly increased spontaneous reversal rates (see below). As a positive control, we tested the response of mutant animals to a high concentration of quinine (10 mM) that that is strongly aversive to wild-type animals. We found that that all strains respond similarly to presentation of 10 mM quinine (S1B Fig). Together, these data show that *inx-18(ok2454)*, *inx-19(ky634)* and *inx-19(tm1896)* mutant animals have increased quinine avoidance, suggesting that ASH activity is increased in the absence of these electrical synapse components.

### The *inx-19(tm1896)* allele alters quinine responses without affecting locomotion

Two different *inx-19* alleles (*tm1896* and *ky634*) have been identified and implicated in sensory neuron function [43]. While mutant animals with either allele show increased response to 1

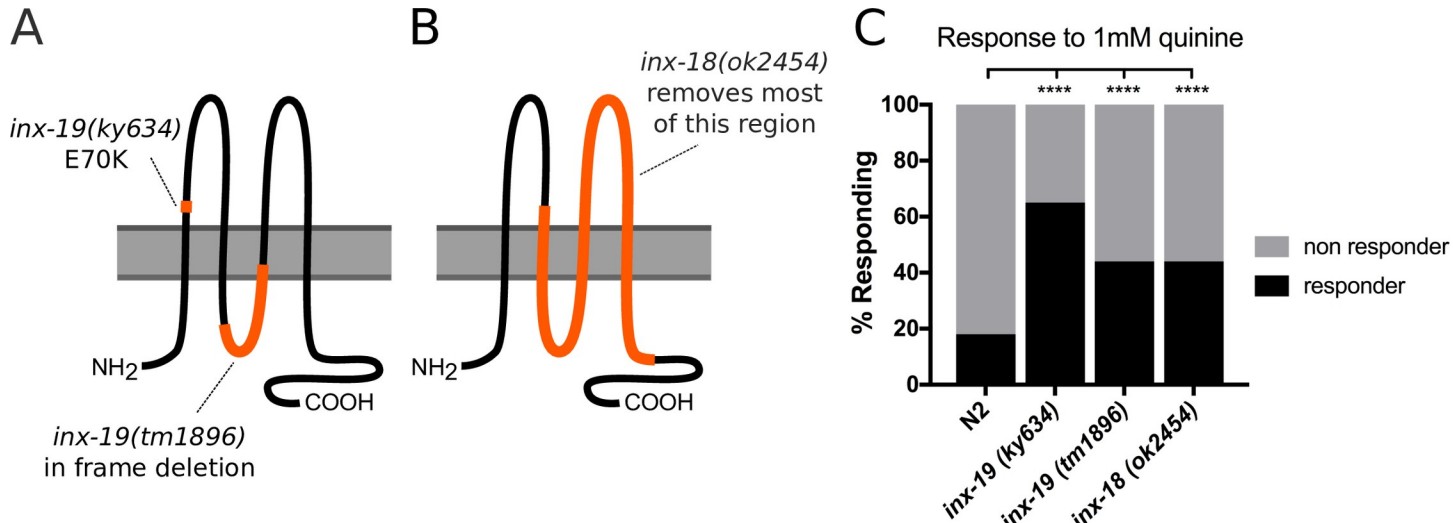

**Fig 1. Mutations in *inx-19* and *inx-18* result in hypersensitivity to quinine.** A,B) Diagram of *inx-19* and *inx-18* alleles used. Innexin genes code for proteins that consist of 4 transmembrane helices with intracellular N and C tails. *Inx-19(ky634)* is a SNP resulting in an E>K substitution within the first extracellular loop, while *inx-19 (tm1896)* is an in-frame deletion of 546bp that removes most of the intracellular loop and a portion of the third transmembrane domain. *Inx-18(ok2454)* is a ~1800bp deletion that removes the second-fourth transmembrane domains and a portion of the C-terminus. C) Quinine Drop Test with 1 mM quinine. *Inx-19(ky634)*, *inx-19 (tm1896)*, and *inx-18(ok2454)* mutant animals are hypersensitive to 1 mM quinine, responding a greater percentage of the time. N2 (wild-type) = 18%, n = 510; *inx-19 (ky634)* = 65%, n = 120, p<0.0001; *inx-19(tm1896)* = 44%, n = 390, p<0.0001; *inx-18(ok2454)* = 44%, n = 350, p<0.0001.

mM quinine (Fig 1C), these two alleles have different impacts on locomotion. First, *inx-19 (ky634)* mutant animals exhibited more reversals in response to M13 (S1A Fig). Second, during locomotion, *inx-19(ky634)* animals spontaneously reversed more frequently in the absence of stimuli (S2A Fig). Third, the average crawling velocity of *inx-19(ky634)* mutant animals was lower than that of wild-type animals (S2B Fig). These data suggest that *inx-19(ky634)* animals have altered movement in addition to changes in quinine response. At a molecular level, *inx-19(ky634)* is a G→A single nucleotide polymorphism causing an E70K substitution within the first extracellular loop of INX-19, while *inx-19(tm1896)* is a 546 basepair deletion that removes the majority of the first intracellular loop and a portion of the second transmembrane domain of INX-19 (Fig 1A). Because the function of innexins requires their transmembrane domains, *tm1896* is likely to be a strong loss-of-function or null allele. By contrast, a substitution within the extracellular docking domain may have a more complicated effect on protein function. For this reason, *inx-19(tm1896)* animals were utilized for the remainder of the experiments.

## *Inx-19* is required in both ASK and ASH for modulation of the quinine response

*Inx-19* is expressed in multiple tissues such as neurons and muscles. Even within the nervous system, *inx-19* is expressed in ASH as well as a number of other neurons, including ASK, which has been implicated in quinine sensation and its regulation [32, 43, 44]. To determine the site of action of INX-19, we performed a series of rescue experiments with *inx-19* cDNA fused to fluorophores in the *inx-19(tm1896)* background. We found that, under the control of the native *inx-19* promoter [43], expression of *inx-19* cDNA fully rescued quinine hypersensitivity in response to 1 mM quinine (Fig 2A). This demonstrates that *inx-19* cDNA is functional and the *inx-19* mutation is responsible for the quinine hypersensitivity phenotype. Interestingly, these worms also showed reduced response to 10 mM quinine, suggesting that INX-19 overexpression could cause over-correction of the quinine sensitivity defects (S3A Fig).

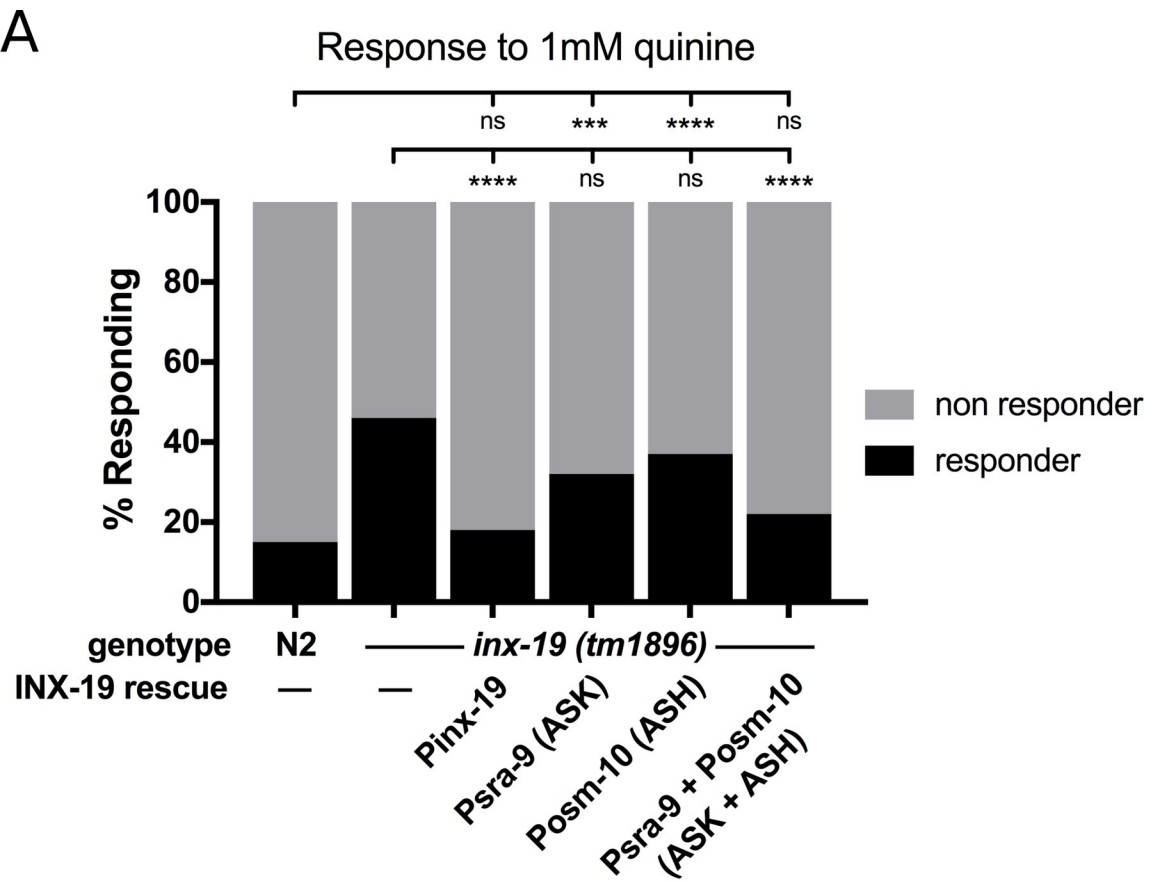

**A** Response to 1mM quinine

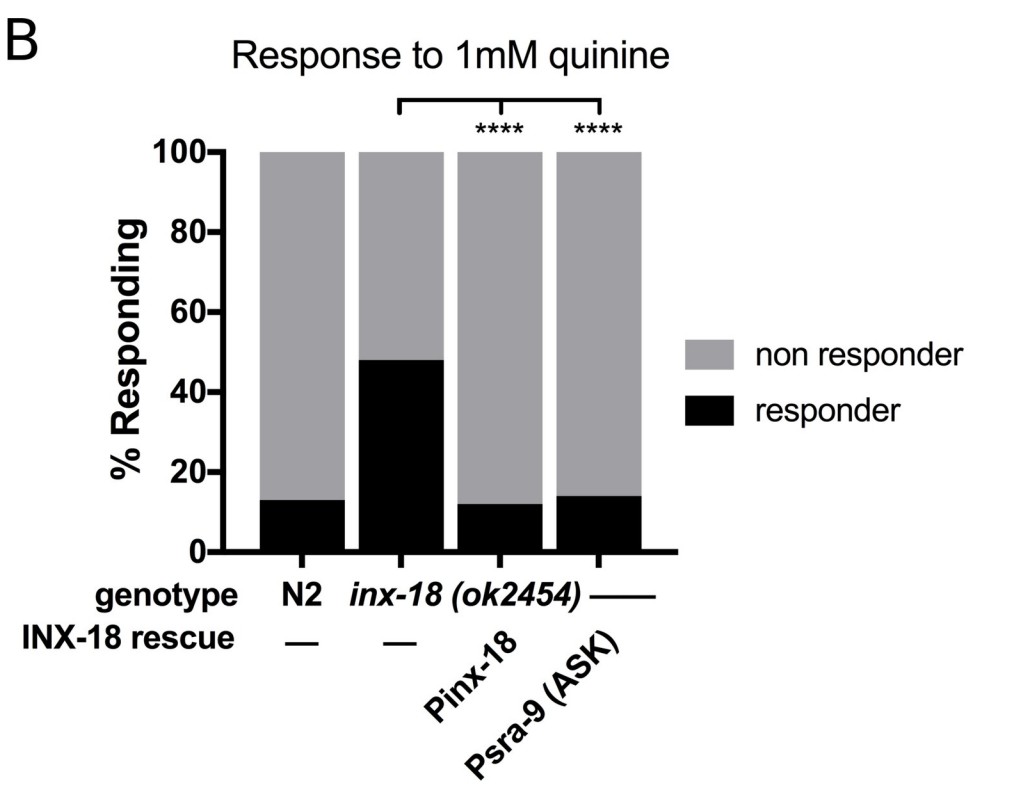

**B** Response to 1mM quinine

**Fig 2. Expression of *inx-19* and *inx-18* in ASK and ASH restores wild-type quinine sensitivity.** A) Expression of *inx-19* isoform A cDNA under the native promoter in *inx-19(tm1896)* animals rescued quinine sensitivity to N2 (wild-type) levels. Expression in ASK (*Psra-9*, which expresses solely in ASK [46]) or ASH (*Posm-10*, which also expresses in the tail neurons PHA and PHB as well as weakly in ASI [47, 48]) alone did not significantly rescue the behavior, while simultaneous expression did. N2 = 15%, n = 220; *inx-19(tm1896)* = 46%, n = 210; *inx-19;Pinx-19::inx-19cDNA* = 18%, n = 100, p = 0.62 vs N2, p<0.0001 vs *inx-19*; *inx-19;Psra-9::inx-19cDNA* = 32%, n = 100, p = 0.0009 vs N2, p = 0.02 vs *inx-19*; *inx-19;Posm-10::inx-19cDNA* = 37%, n = 110, p<0.0001 vs N2, p = 0.13 vs *inx-19*; *inx-19; Psra-9::inx-19cDNA; Posm-10::inx-19cDNA* = 22%, n = 110, p = 0.16 vs N2, p<0.0001 vs *inx-19*. B) Expression of *inx-18* gDNA in *inx-18(ok2454)* animals rescued the quinine hypersensitivity phenotype, as did expression of *inx-18* cDNA in ASK (*Psra-9*). N2 = 13%, n = 120; *inx-18(ok2454)* = 48%, n = 120; *inx-18;inx-18gDNA* = 12%, n = 100, p = 0.84 vs N2, p<0.0001 vs *inx-18*; *inx-18;Psra-9::inx-18cDNA* = 14%, n = 120, p>0.99 vs N2, p<0.0001 vs *inx-18*.

We then expressed GFP or mCherry-tagged *inx-19* cDNA under the control of cell-selective promoters to determine in which neurons INX-19 acts to regulate quinine sensitivity. We found that expression of *inx-19* cDNA in either ASK or ASH (using *Psra-9* [46] and *Posm-10* [47, 48], respectively) did not significantly restore the quinine response to 1 mM quinine in *inx-19(tm1896)* animals. In contrast, simultaneous expression of *inx-19* in both ASK and ASH brought 1 mM quinine response rates back to wild-type levels (Fig 2A). As controls, we tested the response of these animals to M13 buffer and 10 mM quinine and found no change in sensitivity (S3A and S3B Fig). These data indicate that INX-19 is required in both ASK and ASH for appropriate modulation of quinine sensitivity.

## *Inx-18* is required in ASK for modulation of the quinine response

*Inx-18* is expressed in a subset of neurons including ASK [8, 44]. However, unlike *inx-19*, *inx-18* is not expressed in ASH, indicating that its site of action resides outside of ASH. To determine whether the altered quinine response rate of *inx-18* mutant animals is due to the lack of INX-18 function, we performed rescue experiments using *inx-18*. *Inx-18* does not have an obvious promoter, as several genes lie directly upstream of its genomic position. However, the second intron has been successfully used to drive its expression [49]. To test whether the *inx-18(ok2454)* mutation is responsible for the quinine hypersensitivity phenotype, we cloned *inx-18* gDNA, which included the intronic regions. Expression of *inx-18* gDNA was sufficient to restore responses to 1 mM quinine in *inx-18(ok2454)* mutant animals to wild-type levels, indicating that loss of *inx-18* is the reason for quinine hypersensitivity (Fig 2B). Next, we found that the site of action of *inx-18* is in ASK, as expression of *inx-18* cDNA fused to GFP using the *Psra-9* promoter rescued the quinine hypersensitivity phenotype (Fig 2B). As controls, we tested the response of these animals to M13 buffer and 10 mM quinine and found no change in sensitivity (S3C and S3D Fig). These results show that *inx-18* and *inx-19* have distinct, but partially overlapping, sites of action. Combined, our data indicate that INX-19 must be present in both ASK and ASH, while INX-18 in ASK alone is sufficient to modulate the quinine response.

## ASK INX-19 and ASH INX-19 localize to the same regions in neighboring axons

The *C. elegans* wiring diagram suggests that the ASK and ASH neurons form electrical synapses with one another in the nerve ring [41, 42], which raises the possibility that INX-18 and INX-19 are components of these electrical synapses. As our behavioral results show that *inx-19* functions in both ASK and ASH, we examined the subcellular localization of INX-19 in these two neurons using fluorescence microscopy. We drove expression of GFP-tagged INX-19 in ASK and mCherry-tagged INX-19 in ASH. These fluorophore-tagged INX-19 constructs are functional as they can restore quinine responses in *inx-19(tm1896)* mutant animals (Fig 2A). If

INX-19 is a component of electrical synapses between ASK and ASH, we reasoned that INX-19 expressed in ASK would localize to the same regions of the nerve ring as INX-19 expressed in ASH. Our imaging data in wild type animals show that INX-19 forms punctate structures along the axons in the nerve ring when expressed in both cells. As expected, most ASK INX-19 and ASH INX-19 is localized to overlapping puncta, despite the fact that these innexin proteins are in two distinct neurons (Fig 3A–3D). Quantification of these images show that INX-19 expressed in ASK and ASH produces puncta that colocalize 67% of the time (Fig 3H). These data indicate that INX-19 is present on both sides of the ASK-ASH electrical synapses.

## INX-18 rarely colocalizes with INX-19

Our behavioral results indicate that INX-18 functions within ASK to modulate the behavioral response to quinine. To investigate where INX-18 resides in ASK, and whether it is functioning in the same synapses as INX-19, we expressed GFP-tagged INX-18 and asked whether it colocalizes with INX-19 in wild-type animals (Fig3E–3G). We found that, like INX-19, GFP-tagged INX-18 forms puncta along the axons (Fig 3F). However, INX-18 showed low levels of colocalization with mCherry-tagged INX-19 expressed in ASH (~4% colocalization, Fig 3H), demonstrating that the vast majority of INX-18 is not in the same synapses as INX-19 in adult animals.

## INX-19 localization in ASK requires both *inx-18* and *inx-19*

To determine the relationship between INX-18 and INX-19 localization, we investigated whether the expression patterns of INX-18 and INX-19 are influenced by one another. We expressed fluorescently-tagged *inx-18* and *inx-19* cDNA in ASK and ASH individually and examined their expression patterns in mutant backgrounds. We found that the number of INX-19 puncta in the ASK axon was significantly reduced in *inx-18* mutant animals (Fig 4A). In addition, localization of INX-19 within ASK requires INX-19 in other neurons, as the number of ASK INX-19 puncta was diminished in *inx-19(tm1896)* mutant animals (Fig 4A). In no cases were the puncta fully eliminated, indicating that only some electrical synapses are affected in each case. We did not observe significant differences in the number of INX-19 puncta in ASH in *inx-18(ok2454)* or *inx-19(tm1896)* animals, although the downward trend (Fig 4B) suggests that INX-19 localization in ASH may need both *inx-18* and *inx-19*. In contrast, INX-18 localization does not appear to require INX-19, as the number of INX-18 puncta in the nerve ring remained unchanged in *inx-19(tm1896)* mutant animals (Fig 4C). This indicates that the localization of INX-18 is independent of INX-19. Taken together, these data suggest that *inx-18* plays a role in INX-19 electrical synapse assembly and/or maintenance. Perhaps INX-18 is transiently present in the ASK-ASH synapses during development, but by adulthood INX-18 has been removed from these synapses. Indeed, a number of studies have shown that innexin expression can be developmentally contolled [8, 43, 44].

## *Inx-18* and *inx-19* have largely overlapping functions

To investigate the functional relationship between *inx-18* and *inx-19*, we assessed the behavioral responses of *inx-18; inx-19* double mutant animals. If these two genes act in parallel to regulate quinine sensitivity, the phenotype of the double mutant should be stronger than that of the single mutants. If, however, *inx-18* and *inx-19* are acting together in the same pathway, we would expect animals with mutations in both genes to have a phenotype of similar strength to the single mutant animals. The *inx-19(tm1896); inx-18(ok254)* double mutants responded at somewhat higher rates than both the *inx-18(ok2454)* and *inx-19(tm1896)* single mutants (Fig 4D), but this difference was statistically insignificant. This suggests that the two genes function

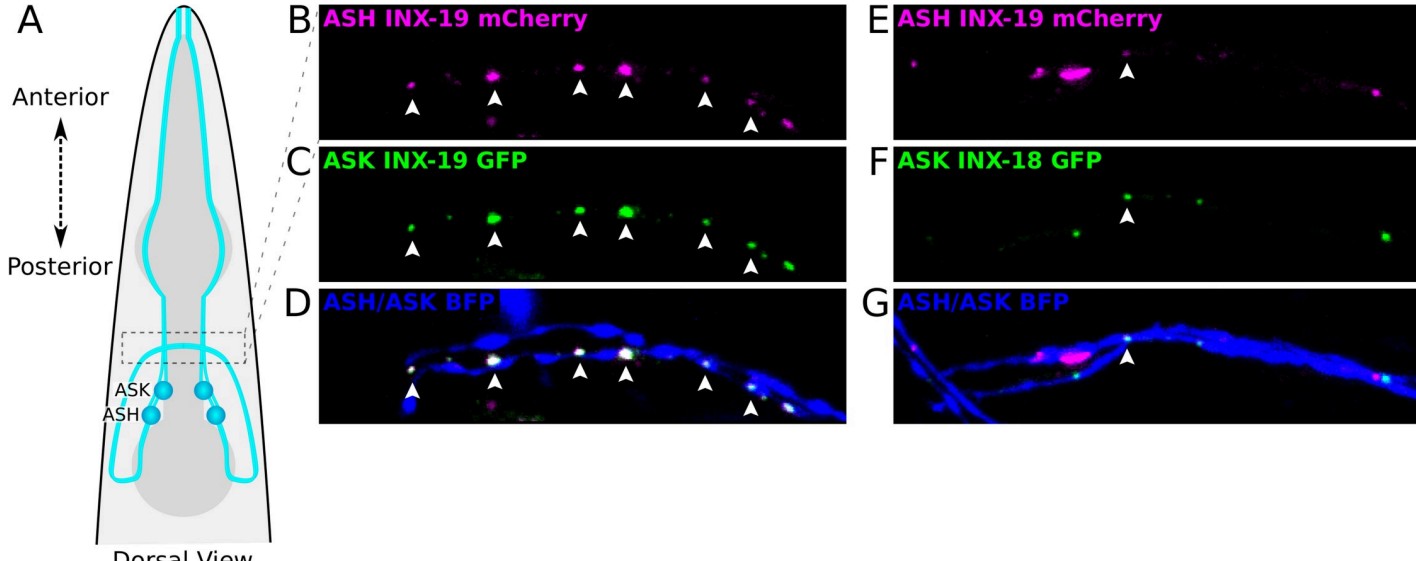

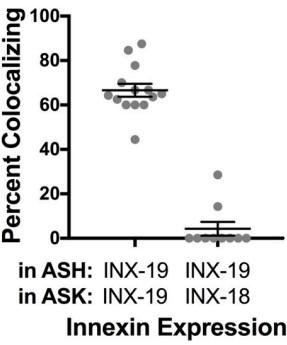

**Fig 3. INX-19 and INX-18 colocalize in the nerve ring when expressed in ASK and ASH.** A) Diagram of the *C. elegans* head in a dorsal view. Dashed box indicates the location of imaging of ASK and ASH axons in the nerve ring. B-D) INX-19 expressed in both ASK (where it is tagged with GFP) (B) and ASH (where it is tagged with mCherry) (C) forms multiple puncta that colocalize along the ASK-ASH axons in wild-type animals. Points of colocalization are indicated with white arrowheads. ASK and ASH are additionally expressing cytosolic mTagBFP2, seen in the axons that traverse the image, highlighted in D. E-G) INX-19 tagged with mCherry expressed in ASH (E) colocalizes in the nerve ring with GFP-tagged INX-18 expressed in ASK in wild-type animals (F). A white arrowhead indicates a point of colocalization. Cytosolic BFP fills the ASK-ASH axons, highlighted in G. H) Quantification of colocalization. In worms expressing INX-19 in ASK and ASH, 67% of nerve ring puncta colocalize (n = 144 puncta in 14 animals). In worms expressing INX-18 in ASK and INX-19 in ASH, ~4% of nerve ring puncta colocalize (n = 81 puncta in 10 animals). Each dot represents an individual worm, and error bars are ±SEM.

largely in the same pathway to modulate the quinine response. Together with the visualization data, these findings suggest that while INX-18 is localized to different electrical synapses than INX-19, its primary function is to set up or maintain INX-19 localization.

### Three different possibilities for the function of the ASK-ASH electrical synapses in quinine regulation

In order to determine how *inx-18* and *inx-19* affect ASH activity, we considered three potential mechanisms: First, *inx-18* and *inx-19* mutations may alter the cell fate of ASK or ASH, leading to changes in the quinine sensing circuit. Second, the ASK-ASH electrical synapses could function to shunt calcium, depressing ASH activity by allowing calcium ions to flow out to ASK. In this case, we expect that removal of ASK-ASH electrical synapses would result in increased

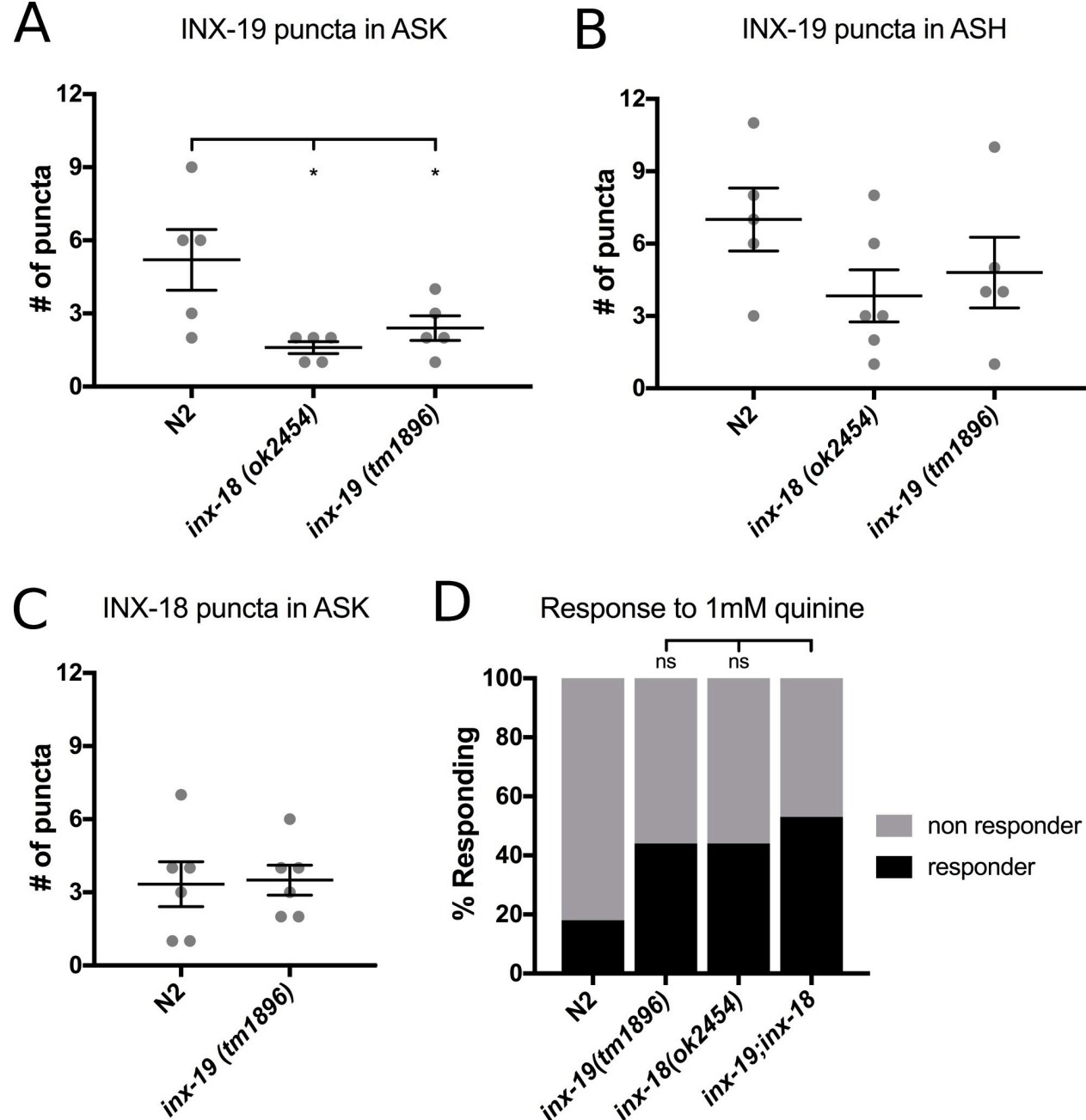

**Fig 4. *inx-18* and *inx-19* play distinct roles in ASK-ASH electrical synapse localization and function.** A) *inx*-19 cDNA was expressed using *Psra-9* and fluorescent puncta in the nerve ring were counted in N2 (wild-type), *inx-18(ok2454)* and *inx-19(tm1896)* backgrounds. Each dot represents an individual worm and error bars are ±SEM. Ordinary one-way ANOVA between three groups showed significant differences (F[2,12] = 5.763, p = 0.02, α = 0.05). Dunnett's multiple comparison test showed that INX-19 ASK puncta were decreased in *inx-18(ok2454)* (n = 5, p = 0.01) and in *inx-19(tm1896)* (n = 5, p = 0.05) in comparison to N2 (n = 5). B) *inx*-19 cDNA was expressed using *Psrd-10* and puncta in the nerve ring were counted in N2, *inx-18(ok2454)* and *inx-19(tm1896)* backgrounds. Each dot represents an individual worm and error bars are ±SEM. Ordinary one-way ANOVA between three groups showed no significant differences (F[2,14] = 0.814, p = 0.46, α = 0.05). C) *inx*-18 cDNA was expressed using *Psra-9* and puncta in the nerve ring were counted in N2, *inx-18(ok2454)* and *inx-19(tm1896)* backgrounds. Each dot represents an individual worm and error bars are ±SEM. Ordinary one-way ANOVA between three groups showed no significant differences (F[2,13] = 1.637, p = 0.23, α = 0.05). D) *Inx-18(ok2454);inx-19(tm1896)* double mutant animals were assayed for sensitivity to 1 mM quinine using the quinine drop test. Double mutants responded at higher rates than either *inx-18* or *inx-19* single mutants. N2 = 18%, n = 510; *inx-19(tm1896)* = 44%, n = 390; *inx-18(ok2454)* = 44%, n = 350; *inx-19;inx-18* = 53%, n = 180, p = 0.05 vs *inx-19*, p = 0.05 vs *inx-18*.

Ca$^{2+}$ signals in ASH and decreased Ca$^{2+}$ levels in ASK. Finally, the ASK-ASH electrical synapses could pass cGMP from ASK to ASH, thus down-regulating the quinine response in ASH. Indeed, it was previously demonstrated that expressing the guanylyl cyclase GCY-27 in ASK rescued the quinine hypersensitivity in *gcy-27(ok3653)* mutant animals [26], suggesting an important role of cGMP in ASK in modulating quinine responses. We tested these three possibilities by examining cell fate markers, the calcium indicator GCaMP6s, and the fluorescent cGMP reporter FlincG3 in ASK and ASH.

## ASK and ASH cell fate and morphology are unchanged in *inx-19* and *inx-18* mutant animals

Electrical synapse channels are known to regulate cell fate decisions during development [50, 51], in particular, *inx-19* has been shown to regulate neural differentiation in *C. elegans* [43]. Thus, it is possible that *inx-19* or *inx-18* also impacts ASK and/or ASH cell fate or morphology. To test this possibility, we expressed mCherry in ASK (using the *sra-9* promoter) and mTagBFP2 in ASH (using the *osm-10* promoter, which also expresses weakly in ASI). We found that the cell fate of ASK and ASH remained the same in the *inx-18(ok2454)* and *inx-19 (tm1896)* mutant animals, as the number of neurons that expressed these fluorescent markers and their positions were unaltered (Fig 5). Furthermore, we showed that the morphology of ASK and ASH were identical between wild-type and the mutant animals. Specifically, both ASK and ASH have cell bodies near the terminal bulb of the pharynx, while dendrites extend to the nose tip and axons project into the nerve ring. Additionally, the cell bodies, dendrites, and axons remained clearly visible In wild-type, *inx-19(tm1896)* and *inx-18(ok2454)* mutant animals (Fig 5B). Together, these data indicate that there is no gross morphological or cell fate changes to either ASK or ASH upon removal of INX-18 and INX-19.

## ASK calcium responses remain unchanged upon removal of ASK-ASH electrical synapses

We examined the possibility that the ASK-ASH electrical synapses function to shunt calcium, thus decreasing behavioral responses to quinine. Previous studies have shown that the ASH neurons respond strongly to quinine with an increase in intracellular calcium [27]. While ASK is known to be a minor player in the quinine response [32], the calcium response of ASK neurons to quinine is unknown. In ASK, attractive stimuli typically result in a decrease in calcium levels, while the aversive stimulus SDS results in a calcium increase [52]. Thus, it is possible that the aversive stimulus quinine also directly triggers a calcium increase in ASK. Alternatively, ASK may receive calcium ions from the primary quinine-sensing neuron ASH via the ASK-ASH electrical synapses. If the ASK-ASH electrical synapses pass calcium from ASH to ASK, this shunting effect would decrease ASH calcium levels in response to quinine as some of the calcium ions in ASH would flow to ASK in wild-type worms. In contrast, in animals lacking the ASK-ASH electrical synapses, we would expect increased calcium levels in ASH as the flow to ASK would be blocked. If ASK receives calcium from ASH, we would expect any quinine-induced calcium signal in ASK to decrease in mutant animals lacking the ASK-ASH electrical synapses.

We expressed GCaMP6s in ASK and ASH to visualize calcium dynamics in those cells in response to quinine presentation. Because both ASK and ASH are involved in blue-light avoidance behavior [53], the GCaMP6s experiments were carried out in a *lite-1(ce314)* background to eliminate blue-light induced changes of GCaMP6s fluorescence in ASK and ASH. Our results showed that CGaMP6 fluorescence in ASK and ASH increased after switching from buffer to quinine, indicating increased Ca$^{2+}$ levels in response to quinine (Fig 6A and 6B, blue

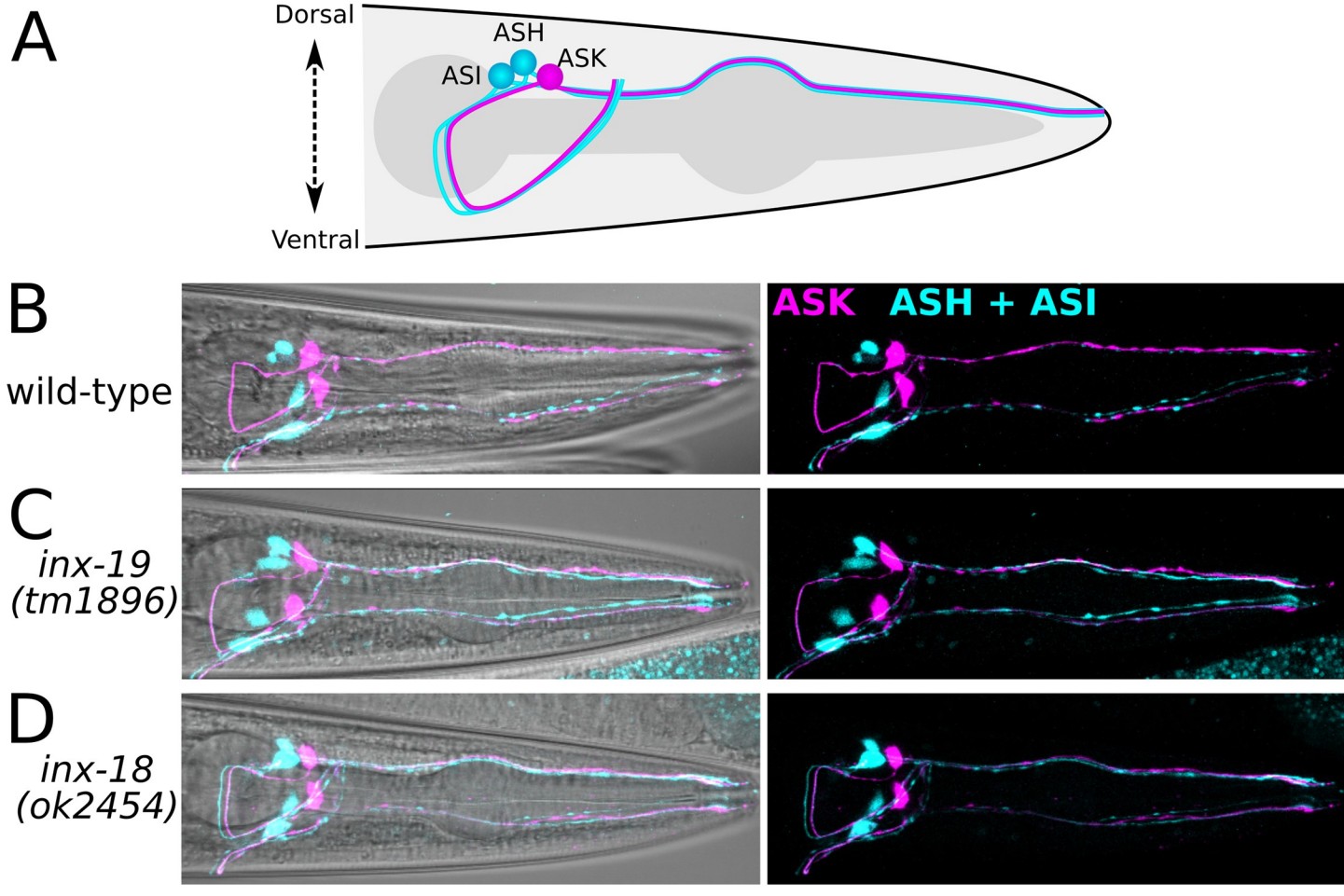

**Fig 5. ASK and ASH architecture is unaltered in *inx-18* and *inx-19* mutant animals.** A) Diagram of neural architecture of ASK, ASH, and ASI in the *C. elegans* head. The dendrites reach out to the nose while the axons extend from the cell body into the nerve ring around the isthmus of the pharynx. B-D) Representative confocal images of the worm head with *Psra-9::mCherry* (ASK) and *Posm-10::bfp* (ASH and weakly in ASI) show cell bodies, dendrites extending to the nose, and axons projecting into the nerve ring. Images on the left include maximum intensity projections of the mCherry and BFP images superimposed upon a brightfield image to show location of cells; images on the right are maximum intensity projections of the mCherry and BFP channels without the brightfield image to show details of the cell architecture. Comparison between wild-type, *inx-19(tm1896)*, and *inx-18(ok2454)* (15–20 animals per genotype were imaged) show no major differences in cell architecture.

traces). However, $Ca^{2+}$ signals in ASH were much more robust than those in ASK, consistent with the role of ASH as the primary quinine-sensing neuron [32].

To examine the impact of electrical synapses on $Ca^{2+}$ dynamics, we monitored ASK and ASH GCaMP6s fluorescence in mutant *inx-18(ok2454)* and *inx-19(tm1896)* animals. We found that the increase in ASK GCaMP6s fluorescence remained the same between wild-type and mutant worms (Fig 6B, 6D and 6F), suggesting that the ASK-ASH electrical synapses are not a main conduit for the ASK $Ca^{2+}$ signal. When we imaged GCaMP6s fluorescence in ASH, we found the increase in ASH GCaMP6s fluorescence were enhanced in *inx-18 (ok2454)* and *inx-19(tm1896)* animals (Fig 6A, 6C and 6E). These results are consistent with the behavioral quinine hypersensitivity observed in these mutant worms. Together, these data show that ASK $Ca^{2+}$ signals do not rely on the ASK-ASH electrical synapses, indicating that $Ca^{2+}$ shunting to ASK is not the primary mechanism of quinine response regulation.

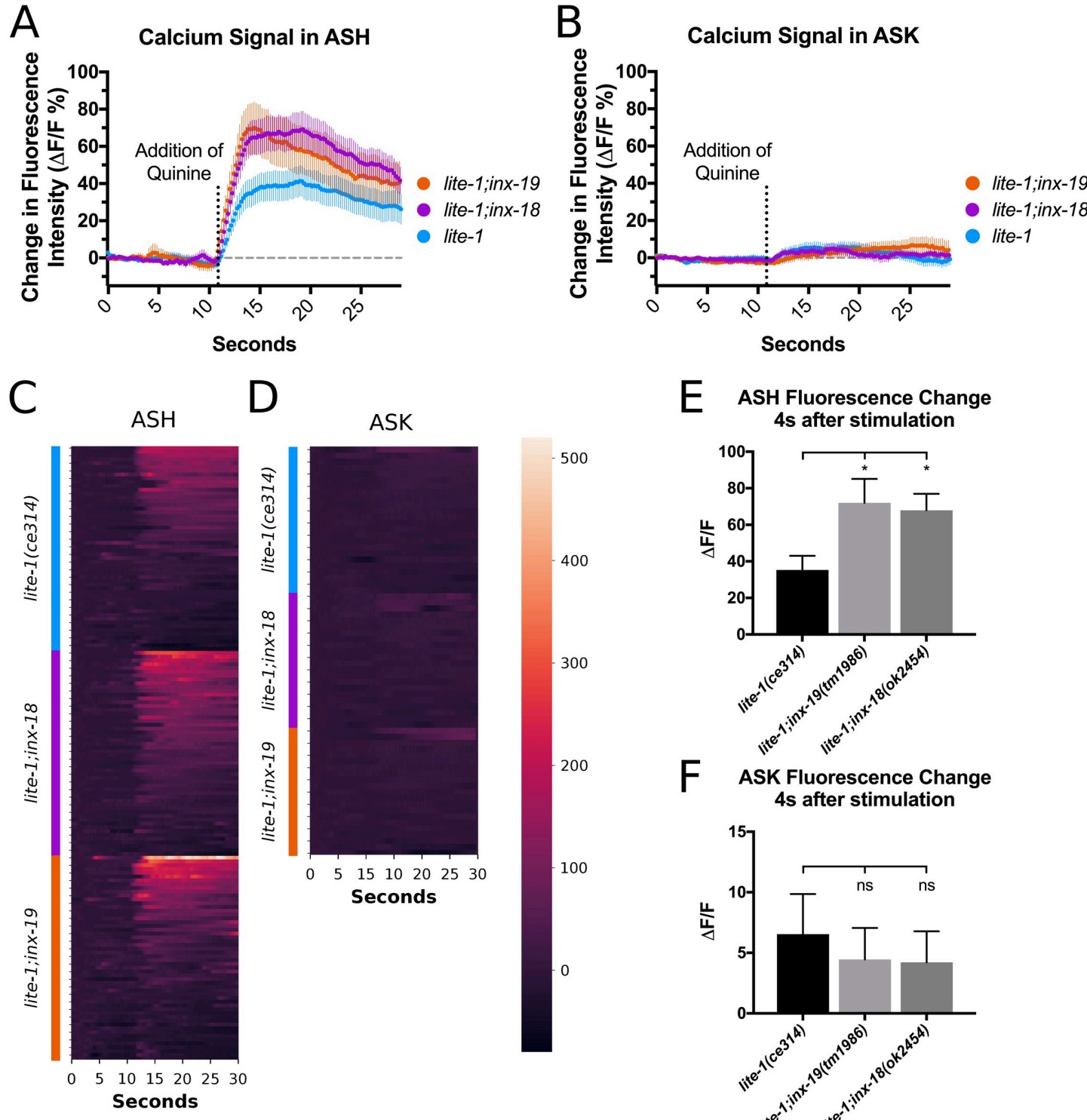

**Fig 6. ASK Ca²⁺ responses to quinine presentation are unaltered in *inx-18* and *inx-19* mutant animals while ASH Ca²⁺ responses are heightened in both.** A)
GCaMP6s fluorescence intensity in ASH in response to 10 mM quinine. Cells were imaged for 30s with presentation of quinine at 10s. The *lite-1(ce314)* mutation was
included to eliminate blue-light induced calcium responses in ASK and ASH. All genotypes showed an increase in ASH GCaMP6s fluorescence in response to quinine
presentation, though for *lite-1;inx-19(tm1896)* and *lite-1;inx-18(ok2454)* animals the response is larger and faster than that of *lite-1(ce314)*. Averaged GCaMP6s traces are
shown and error bars are ±SEM. n = 48 animals for all genotypes tested. B) GCaMP6s fluorescence intensity in ASK in response to 10 mM quinine. ASK showed small
increases of GCaMP6s signals and there were no significant differences between genotypes. Averaged GCaMP traces are shown and error bars are ±SEM. n = 24, n = 21
and n = 22 animals imaged for *lite-1(ce314)*, *lite-1;inx-19* and *lite-1;inx-18*, respectively. C, D) Heatmaps showing individual traces from all worms analyzed. Data points

in the heatmaps represent GCaMP6s signals normalized to the averaged fluorescence intensity of the first 3 seconds of imaging. E) Quantification of ASH fluorescence change at four seconds after quinine stimulation. One-way ANOVA between three groups showed significant differences (F[2,141] = 3.89, p = 0.02, α = 0.05), and Dunnett's multiple comparison test showed that mean ASH GCaMP6s fluorescence change in *lite-1(ce314)* animals (n = 48) differed from both *lite-1;inx-19* (n = 48, p = 0.02) and *lite-1;inx-18* (n = 48, p = 0.05) animals. F) Quantification of ASK fluorescence change four seconds after quinine stimulation. One-way ANOVA between three groups showed no significant differences in ASK GCaMP6s fluorescence (F[2,64] = 0.202, p = 0.817, α = 0.05) between *lite-1(ce314)* (n = 24), *lite-1;inx-19* (n = 21) and *lite-1;inx-18* animals (n = 22).

## cGMP levels in ASK and ASH are influenced by ASK-ASH electrical synapses

cGMP is required within ASH for down regulation of the quinine response [26]. Recently, two studies suggested that guanylyl cyclase expression in other neurons plays a key role in modulating the quinine response [25, 26]. These findings prompted us to examine whether ASH acquires cGMP through the ASK-ASH electrical synapses. Indeed, ASK expresses the guanylyl cyclases ODR-1 and GCY-27 [54], both of which are known to modify the quinine response [25, 26]. If ASK supplies ASH with cGMP through the ASK-ASH electrical synapses, we would expect to observe diminished levels of cGMP in ASH with a compensatory increase within ASK in *inx-18(ok2454)* and *inx-19(tm1896)* mutant animals.

To visualize levels of cGMP within ASK and ASH, we utilized the *C. elegans* codon-optimized version of FlincG3, which contains the cGMP binding domains of protein kinase G1α fused to cpEGFP [55, 56]. Binding of cGMP increases FlincG3 fluorescence. We co-expressed FlincG3 and the red fluorescent protein mScarlet under control of the same promoters in ASK and ASH in the *lite-1(ce314)* background (Fig 7A). After crossing the transgenes into *inx-18 (ok2454)* and *inx-19(tm1896)*, we imaged FlincG3 fluorescence in ASK and ASH. FlincG3 fluorescence was compared to mScarlet fluorescence to account for variations in expression levels. We found that ASH FlincG3 fluorescence was decreased in both *inx-18(ok2454)* and *inx-19 (tm1896)* mutant animals (Fig 7B), suggesting a reduction of the basal cGMP levels in ASH. These data are consistent with the behavioral hyper-responsiveness of *inx-18* and *inx-19* mutant worms to dilute quinine, as decreased cGMP levels could lead to increased ASH calcium levels in response to quinine [25, 26]. In ASK, FlincG3 fluorescence was increased in *inx-19(tm1896)* mutant animals but was unchanged in *inx-18(ok2454)* animals (Fig 7C), suggesting that INX-19-based electrical synapses are primarily responsible for supplying ASH with cGMP from ASK. Together, our data suggest that INX-18 and INX-19 are major components of the ASK-ASH electrical synapses that modulate behavioral sensitivity to quinine, and that they do so by affecting transport of cGMP into ASH.

## Discussion

We showed that electrical synapses between the *C. elegans* sensory neurons ASK and ASH play an active role in modifying nociceptive behavior via the passage of cGMP between cells. We found that the innexins INX-18 and INX-19 are required within ASK and ASH for proper modulation of the quinine response, as mutant animals lacking these innexins are hyperresponsive to quinine. These innexins form electrical synapses between ASK and ASH, in which INX-19 is a major component, though INX-18 is important for correct localization of INX-19 synapses in ASK. Our study supports a model in which ASK-ASH electrical synapses facilitate the passage of cGMP from ASK to ASH. Within ASH, cGMP downregulates calcium signals in response to quinine stimulation, likely by binding to and activating the cGMP-dependent protein kinase EGL-4 [26], ultimately leading to a reduction neural activity and thus aversive behavior (Fig 8).

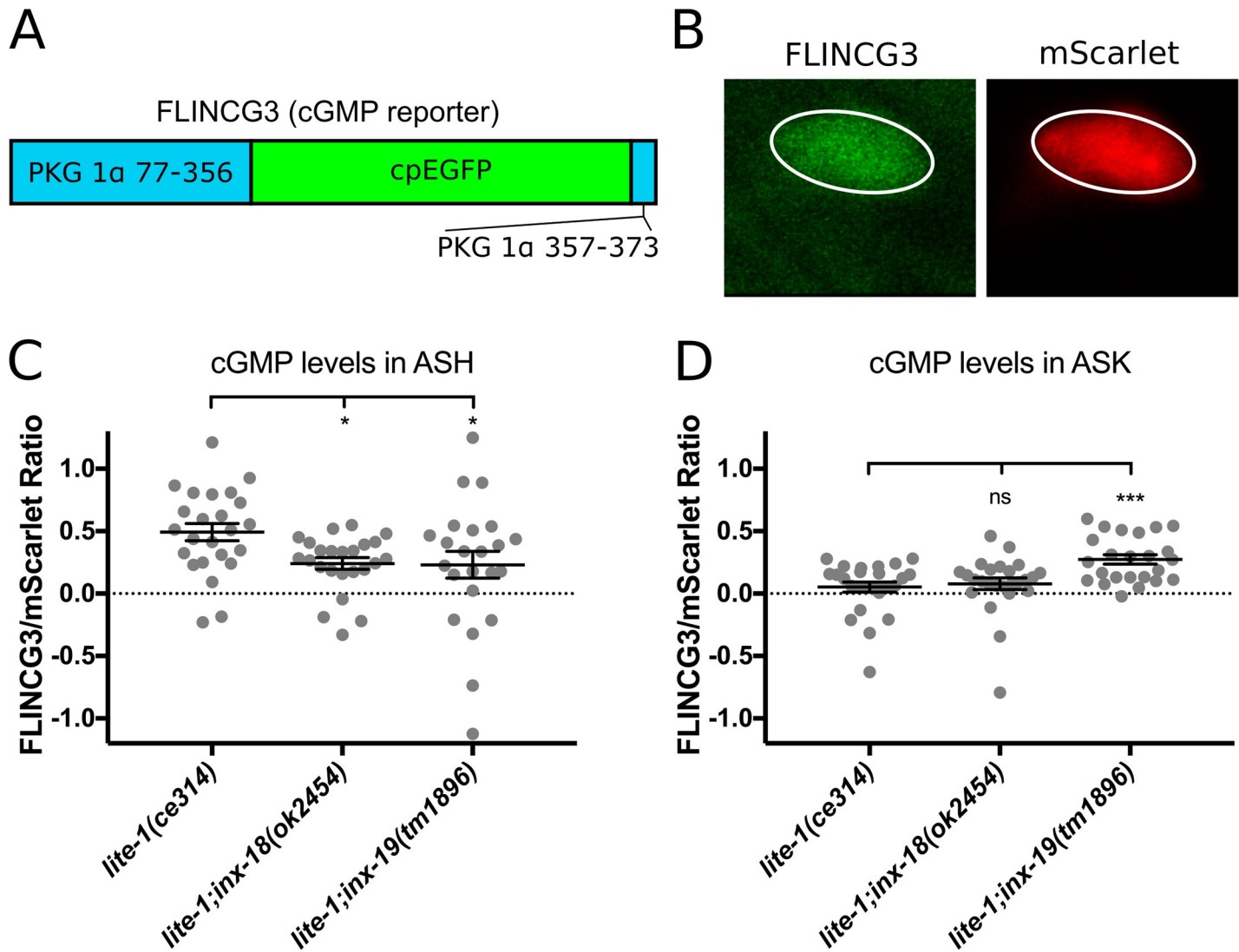

**Fig 7. Mutations in *inx-18* and *inx-19* disrupt endogenous cGMP levels in ASK and ASH.** A) Diagram of FlincG3. The cGMP binding domains of PKG 1α (blue) are followed by circularly permuted EGFP (green) and a short PKG 1α tail (blue). WingG2 increases in brightness in response to cGMP. B) Example of FlincG3 and mScarlet expression within ASH. Ellipses were drawn around the cell body to measure fluorescence intensity. C) cGMP levels within the ASH cell body. The ratio between mean fluorescence intensity of FlincG3 and mScarlet signals was determined for each genotype. Decreases in ASH FlincG3 fluorescence were found in *inx-18 (ok2454)* and *inx-19(tm1896)* mutant animals when compared to wild-type worms. Each data point was obtained from a single cell; error bars are ±SEM. One-way ANOVA between three groups showed significant differences (F[2,68] = 3.643, p = 0.03, α = 0.05), and Dunnett's multiple comparison test showed that mean fluorescence intensity in *lite-1(ce314)* (n = 24) cells differed from both *lite-1;inx-18* cells (n = 24, p = 0.05) and *lite-1;inx-19* cells (n = 23, p = 0.04). D) cGMP levels within the ASK cell body. ASK FlincG3 fluorescence was not altered in *inx-18(ok2454)* mutant animals, and increased in *inx-19(tm1896)* mutant animals when compared to wild-type animals. Each data point was obtained from a single cell; error bars are ±SEM. One-way ANOVA between three groups showed significant differences (F[2,72] = 8.115, p = 0.0007, α = 0.05), and Dunnett's multiple comparison test showed that mean fluorescence intensity in *lite-1(ce314)* cells (n = 26) did not differ from *lite-1;inx-18* cells (n = 25, p = 0.87) but was increased in *lite-1;inx-19* cells (n = 24, p = 0.0008).

Our study supports a model in which ASK-ASH electrical synapses facilitate the passage of cGMP from ASK to ASH. Within ASH, cGMP downregulates calcium signals in response to quinine stimulation, leading to a reduction in aversive behavior. INX-19 (orange) is shown on both sides of the ASK-ASH electrical synapses while INX-18 (purple) is shown joining with an unknown innexin and contributing to INX-19-based synapse localization.

Electrical synapses can be made of different combinations of innexin subunits. Homotypic channels contain hemichannels that are composed of the same innexins, while heterotypic

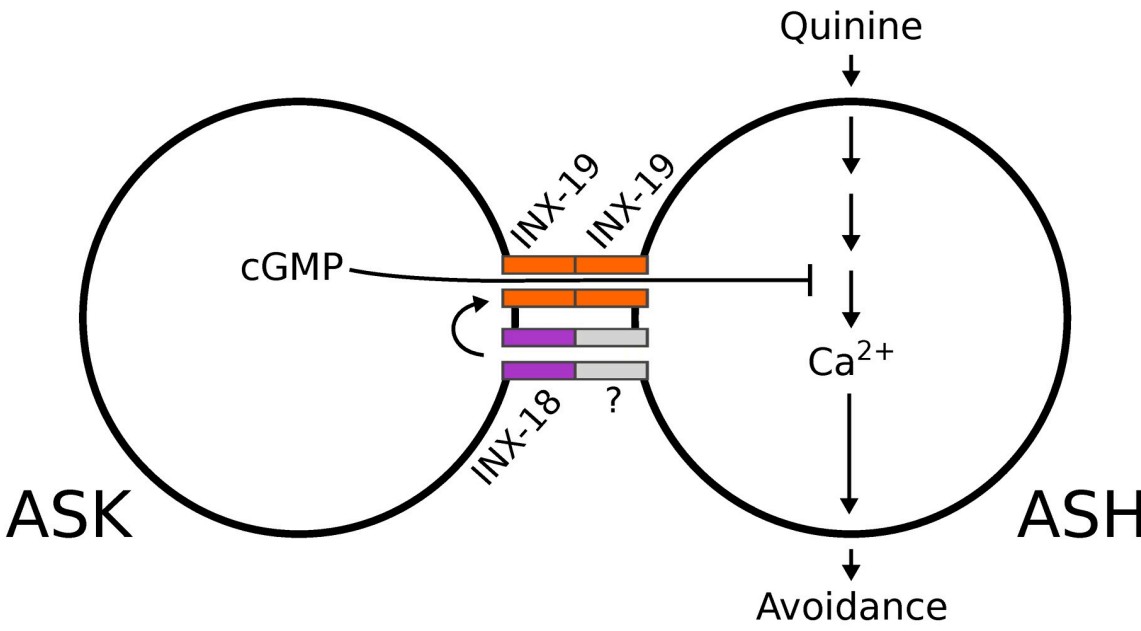

**Fig 8. Model of ASK-ASH electrical synapse facilitation of ASH modulation.**

channels are made up of hemichannels that are composed of different innexins. The channel composition determines permeability, as heterotypic channels are thought to produce rectified electrical synapses: those that preferentially pass ions and small molecules in one direction rather than equally in both [57–59]. Our data suggest that INX-19 is a major component of the ASK-ASH electrical synapses. One possibility is that INX-19 forms homotypic channels. However, some INX-19 synapses do contain INX-18, suggesting that at least some are heterotypic. Though the number of electrical synapses containing both INX-18 and INX-19 is quite small, it is possible that levels of INX-18 within such synapses are generally low, making their visualization difficult. INX-18 could also make electrical synapses with other innexins in ASH. Nonetheless, our results suggest that the main function of INX-18 is carried out through its regulation of INX-19, as the *inx-18* and *inx-19* mutants do not show additive responses to 1mM quinine.

The structural makeup of the ASK-ASH electrical synapses has functional implications for ASH modulation. The composition of electrical synapses is key in determining their permeability of small molecules (such as $Ca^{2+}$ and cGMP), and heterotypic composition is a major cause of rectification [57, 59–61]. If the ASK-ASH electrical synapses are heterotypic (*i.e.*, consist of both INX-18 and INX-19 hemichannels) and rectified, this could explain why ASK cGMP levels, but not calcium levels, are affected by *inx-18* and *inx-19* mutations. Rectified channels bias the direction of movement of ions and molecules, making it more likely for signals to travel in one direction. If small molecule signals could easily pass from ASK to ASH but not in the reverse direction, cGMP may be more likely to travel from ASK to ASH than $Ca^{2+}$ would be from ASH to ASK. This mechanism could explain why our data suggest movement of cGMP but not $Ca^{2+}$. Additionally, the permeability of electrical synapses is dependent on the subunits that make up the channels [17, 62]. While the permeability of most innexin-based channels is unknown, it is possible that the ASK-ASH electrical synapses are more permeable to cGMP than $Ca^{2+}$, particularly given the timescales upon which each operate. Electrical synapses have long been considered low-pass filters, preferentially passing signals that change over longer time periods as opposed to quick oscillations [63, 64]. Regardless of the molecular

reason, the selectivity of electrical synapses to either particular molecules or directions means that they can be sophisticated players within neural circuits. Changes in innexin composition during development or in mature circuits could dramatically impact how the neurons are regulated through the electrical synaptic network.

Electrical synapses are not static structures; they are regulated developmentally as well as in mature circuits [44, 63, 65–67]. Our data suggest that innexins can impact the localization of other innexins even if they are not a permanent part of the same synapses. INX-18 plays a crucial role in the localization of INX-19. Thus, its main impact on modulating the quinine response may be in supporting the function of INX-19. While INX-18 is required for proper localization of INX-19, an *inx-18* mutation does not eliminate INX-19 synapses completely. This may explain why the *inx-18(ok2454)* mutation does not have an impact on cGMP levels in ASK, as some signaling could still occur through the remaining INX-19-based electrical synapses even in the absence of INX-18.

ASH activity is modulated by cGMP, and yet ASH is not known to express any guanylyl cyclases, which produce cGMP [54, 68, 69]. This suggests that other neurons may regulate its activity. Such modulation occurs in the context of a larger sensory neuron network that simultaneously assesses many different sensory inputs, any of which could be affecting baseline levels of cGMP within sensory neurons. Thus, by being sensitive to changes in cGMP levels, ASH is able to receive modulatory information from many neurons simultaneously. ASH receives cGMP from its immediate neighbor ASK as well as other neurons [25], suggesting that cGMP levels within ASH (and thus nociceptive sensitivity) are under the control of a number of external signals. If this is the case, cGMP could be a general signal of the state of the worm, integrating multiple signals to indicate whether it is in a favorable or unfavorable circumstance [70–74]. Our data support the notion that electrical synapses regulate function in a sensory neuron network by modulating the passage of small molecules into neurons such as ASH. In this way, multiple sensory inputs such as availability of food or sexual partners, presence of pathogens or other environmental conditions could alter various different behaviors at once.

## Materials and methods

### *C. elegans* culture

Strains were maintained at room temperature (20–21˚C) on NGM agar plates seeded with OP50 *E. coli* bacteria. The N2 strain (Bristol, England) was used as wild type. The following mutant strains were used in this study: CX6161 *inx-19 (ky634) I*, FX01896 *inx-19 (tm1896) I*, RB1896 *inx-18 (ok2454) IV*, BJH2183 *inx-18 (ok2454) IV;inx-19(tm1896) I*, BJH2259 *lite-1 (ce314) X*, BJH2304 *lite-1(ce314);inx-19(tm1896)*, and BJH2303 *lite-1(ce314);inx-18(ok2454)*.

### Transgenes

Transgenic strains for rescue experiments were generated by microinjection [75] of various innexin-containing plasmids (30–40 ng/μl) together with co-injection markers (Table 1). The co-injection markers were *Punc-122*::*gfp* (BJP-I15, 20–40 ng/μl) and *Punc-122*::*mcherry* (BJP-I14, 30–40 ng/μl). Cytoplasmic fluorophores (mCherry, mTagBFP2, and mScarlet) were injected at 30-40ng/μl. For GCaMP imaging experiments, plasmids (BJP-L136, *Psrbc-66*:: *GCaMP6s*::*SL2*::*mCherry*::*let-858utr* or BJP-L137, *Posm-10*::*GCaMP6s*::*SL2*::*mCherry*::*let-858utr*) were injected at 70 ng/μl into the light-insensitive *lite-1(ce314)* worms. To quantify cGMP levels, FlincG3 plasmids (pFG270, *Psrbc-66*::*FlincG3*::*unc-54utr* or pFG250, *Psrd-10*:: *FlincG3*::*unc-54utr*) were injected at 20 ng/μl into *lite-1(ce314)* worms.

**Table 1. DNA constructs.**

| Name | Construct | Construction Notes |
|------|-----------|--------------------|
| BJP-L109 | *Pinx-19::inx-19a::gfp::unc-54utr* | *Pinx-19* (5556bp) is from Dr. Cornelia Bargmann and primers were: GATAAGCGCGGATGCTCCT and TGACAGTGCTCTCAGAGGGA.<br>*Inx-19a* cDNA is from Dr. Cornelia Bargmann and primers were: ATGTGGCGGACACCAGCATC and AAGAAACGATTTCGTCTGTCCAGGA. |
| BJP-I15 | *Punc-122::gfp::unc-54utr* | |
| BJP-L99 | *Psra-9::inx-19a::mCherry::gdp-2utr* | *Psra-9* is 3012bp and primers were: GCATGCTATATTCCACCAAA and GAAATCTTGAAACTGAAAAATACA |
| BJP-L112 | *Psra-9::inx-19a::gfp::unc-54utr* | |
| BJP-L125 | *Psra-6::inx-19a::mCherry::gdp-2utr* | *Psra-6* is 2018bp and primers were TTCCAGTGCTCTGAAAATCTTG and GGCAAAATCTGAAATAATAAATATT |
| BJP-L114 | *Posm-10::inx-19a::gfp::unc-54utr* | *Posm-10* (900bp) is from Dr. Josh Kaplan and primers were: CTTGACACCGACTGGCAC and GCGTTCGACACCTTGTAAGAT |
| BJP-L120 | *Psrd-10::inx-19a::gfp::unc-54utr* | *Psrd-10* (1841bp) is from Dr. Denise Ferkey and primers were: AGCCACGGCTAGCTACAG and GTTGAATTTGGTCTGTGAGCT |
| | inx-18 gDNA PCR | *Inx18* gDNA (7646bp) used the primers: ACAGTCGAGTCGTCGTCGTCG and TAATTTTGAAACAAAAATCGGAAAGAA |
| BJP-L46 | *Psra-9::inx-18::gfp::unc-54utr* | *Inx-18* cDNA (1308bp) is from Dr. Zhao-Wen Wang and primers were: ATGGTCGGTGGATTCCG and AACATAATGTGTCCGTGTCGGA |
| BJP-L115 | *Psrbc-66::mTagBFP2::unc-54utr* | *Psrbc-66* is 2055bp and used the primers: CAACGATGAAATATTGATCGTACAAA and TTCTGAGACACCTGACTTTCTGTC |
| BJP-L116 | *Posm-10::mTagBFP2::unc-54utr* | |
| BJP-L143 | *Psrbc-66::mScarlet::unc-54utr* | |
| BJP-L142 | *Psrd-10::mScarlet::unc-54utr* | |
| BJP-L139 | *Psra-9::mCherry::unc-54utr* | |
| BJP-L136 | *Psrbc-66::GCaMP6s::SL2::mCherry::let-858utr* | |
| BJP-L137 | *Posm-10::GCaMP6s::SL2::mCherry::let-858utr* | |
| pFG270 | *Psrbc-66::FlincG3::unc-54utr* | Received from Dr. Denise Ferkey |
| pFG250 | *Psrd-10::FlincG3::unc-54utr* | Received from Dr. Denise Ferkey |

## Behavioral assays

Well-fed day 1 adults were used for all analyses. To ensure uniformity of worm age and feeding status, L4 animals were picked onto fresh plates the afternoon before behavior tests. Behavior assays were performed on at least 5 separate days in parallel with controls.

## Quinine drop test

The quinine drop test was performed as described previously [31, 32, 45]. Quinine HCl (Sigma-Aldrich Q1125) was dissolved in M13 Buffer pH 7.4 (30 mM Tris-HCl pH 7.0, 100 mM NaCl, 10 mM KCl) to 10 mM. Aliquots were frozen at -20°C. Aliquots were defrosted on the day of the experiment and allowed to reach room temperature before use. Solutions were loaded into glass needles via mouth pipetting through long silicone tubing. Needles were formed from 1.5 mm filamented glass capillaries (World Precision Instruments, Inc.) with a Sutter micropipette puller and the tips opened by breaking with fine forceps. 10cm NGM plates were brought to room temperature on the bench overnight and then left open at room temperature to dry for 2.5–4 hours before being used (plates are appropriately dry when worms leave tracks on the agar that do not immediately disappear). For each assay, 15 worms were placed on a plate and allowed to acclimate for 30 min. Small drops (approximately 1 body length in diameter) of M13, 1 mM quinine, or 10 mM quinine were then delivered via glass needle approximately 1 body length in front of worms. When worms encountered the

drop, they were scored as avoiding the drop if they initiated a reversal within 4 s and reversed at least half a body length backwards. To avoid desensitization, worms were not tested with a new solution within 2min of initial drop presentation. The experimenter was blind to the strain when scoring reversals. All rescues except for *inx-18* gDNA were performed with C-terminal mCherry- or GFP-tagged INX-19 or INX-18 and expression was verified visually before behavioral experiments. All groups were compared with a Chi-square test and post-hoc Fisher's Exact tests with Bonferroni's correction were computed to compare groups

## Movement assays

Five worms at a time were placed on 10 cm NGM plates and allowed to acclimate for 1 minute. Video capturing was then carried out using an imaging set up from MBF Bioscience. Freely crawling worms were monitored for 60 seconds at 5 frames per second. Moving velocity at each frame was calculated by the WormLab 4.1 from MBF Bioscience after confirming correct assignment of head location throughout the video. Reversals were denoted with negative values. Comparison of number of reversals/min and mean velocity was calculated using an ordinary one-way ANOVA using Dunnett's correction for multiple comparisons between all groups. The alpha value was set at 0.05.

## Confocal microscopy for imaging synapse and cell architecture

Young adults were paralyzed using 30 mg/ml 2,3-butanedione monoxime (BDM) dissolved in M9. Worms were imaged using an Olympus FV1000 confocal system with a 60x oil objective (NA 1.4). Z-stacks of fluorescent images (0.40 μm step-size for synapses, or 1.20 μm step-size for cell architecture) were taken at the region of interest. Maximum intensity projections of images were obtained using Fiji. For colocalization analysis, mTagBFP2 was cytoplasmically expressed in ASK and ASH in order to visualize axons. The number of INX-18 and INX-19 puncta within mTagBFP2-expressing ASH and ASK axons in the nerve ring was counted. Puncta were scored as colocalizing (containing signal from both channels) or non-colocalizing (containing signal from a single channel). Percentage colocalization was calculated by determining the ratio between the number of colocalizing puncta and the total number of puncta in each maximum intensity projection.

## Calcium imaging

GCaMP6s [76] was used for all calcium imaging. *Lite-1(ce314)* worms were injected with either *Psra9::GCaMP6s::SL2::mCherry::let-858utr* (ASK) or *Posm-10::GCaMP6s::SL2::mCherry::let-858utr* (ASH) along with the co-injection marker *Punc-122:mCherry*. Transgenic lines were crossed with mutant animals to generate *lite-1(ce314);inx-19(tm1896)* and *lite-1(ce314);inx-18 (ok2454)*, which the identical extrachromosomal arrays for imaging. Worms were imaged using a microfluidic olfactory chip [77]. M13 buffer was used to load worms into the chip, and their nose tips were washed with M13 buffer for 30 seconds before each recording. At the start of the recording, animals were exposed to M13 buffer for 10 s before 10 mM Quinine dissolved in M13 was washed in to the chip. The images were captured at 5 frames per second with an exposure time of 100ms on a Leica DMI3000B inverted microscope with a 63x Oil objective and a QImaging OptiMOS camera. The region of interest was defined as a square-shaped area surrounding the desired cell body. Background-subtracted fluorescence intensity values were collected from every sample's ROI and stored into MATLAB formatted files. Change in fluorescence intensity (ΔF/F%) was calculated by dividing each value by the average intensity of the first 3 seconds of imaging.

## cGMP imaging

FlincG3 [55, 56] was used for cGMP imaging. *Lite-1(ce314)* worms were injected with either *Psrbc-66::FlincG3::unc-54utr* and *Psrbc-66::mScarlet::unc-54utr* (ASK) or *Psrd-10::FlincG3::unc-54utr* and *Psrd-10::mScarlet::unc-54utr* (ASH) along with the co-injection marker *Punc-122:mCherry*. Transgenic lines were crossed with mutant animals to generate *lite-1(ce314);inx-19(tm1896)* and *lite-1(ce314);inx-18(ok2454)*, which carry the identical extrachromosomal arrays for imaging. L4 worms were picked onto fresh OP50-seeded NGM plates 6 hours before imaging to ensure synchronization of age and feeding status. Young adults were paralyzed with 30 mg/ml BDM dissolved in M9. Immobilized worms were imaged using an Olympus FV1000 confocal microscope with a 60x Water objective. Kalman filtering was used to reduce noise. Z-stacks (1.28 μm step-size) were taken through the cell body. Maximum intensity projections were obtained using Fiji [78]. Two elliptical ROIs were drawn in the mScarlet channel: one surrounding the cell body and one capturing background fluorescence from a region near the cell body that did not contain an axon or dendrite. Mean pixel intensity in both the FlincG3 and mScarlet channels was calculated using Fiji and background intensity was subtracted from cell body intensity. The ratio between FlincG3 and mScarlet mean intensity was calculated to control for expression variation.

## Statistical analyses

Statistical analyses for all experiments except calcium imaging were carried out as described in the legends for each figure using GraphPad Prism Statistical analysis of the calcium imaging experiments was carried out using a custom written MATLAB program and GraphPad Prism.

## Supporting information

**S1 Fig. *inx-18* and *inx-19* mutant animals respond normally to control solutions. A)** *Inx-19(tm1896)* and *inx-18(ok2454)* mutant animals respond at N2 (wild-type) levels when presented with M13 buffer, while *inx-19(ky634)* animals respond slightly more than wild-type animals. N2 = 13%, n = 330; *inx-19(ky634)* = 23%, n = 120, p = 0.012; *inx-19(tm1896)* = 19%, n = 210, p = 0.07; *inx-18(ok2454)* = 16%, n = 160, p = 0.33. **B)** *Inx-19(ky634)*, *inx-19(tm1896)*, and *inx-18(ok2454)* mutant animals respond at wild-type levels when presented with 10 mM quinine. N2 = 93%, n = 330; *inx-19(ky634)* = 97%, n = 120, p = 0.18; *inx-19(tm1896)* = 97%, n = 210, p = 0.03; *inx-18(ok2454)* = 98%, n = 120, p = 0.02.
(TIF)

**S2 Fig. *inx-19(ky634)* mutant animals have movement defects. A)** *Inx-19(ky634)* mutant animals reverse more frequently than N2 (wild-type) animals. Number of reversals were counted from a one-minute video. One-way ANOVA between three groups showed significant differences ($F_{[2,99]}$ = 6.943, p = 0.0015, α = 0.05), and Dunnett's multiple comparison test showed that N2 (n = 34) differed from *inx-19(ky634)* (n = 33, p = 0.0006) but not *inx-19(tm1896)*(n = 35, p = 0.097). **B)** *Inx-19(ky634)* mutant animals have lower average movement velocity than N2 animals. One-way ANOVA between three groups showed significant differences ($F_{[2,99]}$ = 6.089, p = 0.003, α = 0.05), and Dunnett's multiple comparison test showed that N2 (n = 34) differed from *inx-19(ky634)* (n = 33, p = 0.021) but not *inx-19(tm1896)* (n = 35, p = 0.677). Each data point represents a single worm and error bars are ±SEM.
(TIF)

**S3 Fig. Responses of worms carrying rescue transgenes to negative and positive control solutions. A)** *Inx-19(tm1896)* animals carrying rescue transgenes behaved like N2 (wild-type) animals when presented with M13 buffer. N2 = 14%, n = 220; *inx-19(tm1896)* = 19%, n = 210;

*inx-19;Pinx-19::inx-19cDNA* = 10%, n = 100; *inx-19;Psra-9*::*inx-19cDNA* = 10%, n = 100; *inx-19;Posm-10::inx-19cDNA* = 11%, n = 110; *inx-19;Psra-9*::*inx-19cDNA; Posm-10::inx-19cDNA* = 10%, n = 110. **B)** *Inx-18(ok2454)* animals carrying rescue transgenes behaved like N2 animals when presented with M13 buffer. N2 = 12%, n = 120; *inx-18(ok2454)* = 7%, n = 120; *inx-18; inx-18gDNA* = 4%, n = 100; *inx-18;Psra-9*::*inx-18cDNA* = 9%, n = 120. **C)** *Inx-19(tm1896)* animal carrying neuron-specific transgenes behaved like N2 animals when presented with 10 mM quinine, but expression of *inx-19* cDNA using the native promoter reduced the responses to 10 mM quinine below wild-type levels. N2 = 96%, n = 220; *inx-19(tm1896)* = 97%, n = 210; *inx-19;Pinx-19::inx-19cDNA* = 85%, n = 100, p = 0.002 vs N2, p = 0.0004 vs *inx-19*; *inx-19; Psra-9*::*inx-19cDNA* = 91%, n = 100, p = 0.10 vs N2, p = 0.04 vs *inx-19*; *inx-19;Posm-10::inx-19cDNA* = 97%, n = 110, p = 0.76 vs N2, p>0.99 vs *inx-19*; *inx-19;Psra-9*::*inx-19cDNA; Posm-10::inx-19cDNA* = 96%, n = 110, p>0.99 vs N2, p = 0.74 vs *inx-19*. **D)** When expressing *inx-18* cDNA under the native promoter or in ASK, *inx-18(ok2454)* animals behaved like wild-type when presented with 10 mM quinine. N2 = 97%, n = 120; *inx-18(ok2454)* = 95%, n = 120; *inx-18;inx-18gDNA* = 91%, n = 100; *inx-18;Psra-9*::*inx-18cDNA* = 91%, n = 120. (TIF)

**S1 Table. Archive of raw data for figures.**
(XLSX)

## Acknowledgments

The authors thank the *Caenorhabditis elegans* Genetic Consortium (funded by NIH Office of Research Infrastructure Programs P40 OD010440) and Dr. Shohei Mitani for worm strains; Dr. Cori Bargmann, Dr. Josh Kaplan, and Dr. Zhao-Wen Wang for plasmids.

## Author Contributions

**Conceptualization:** Lisa Voelker, Ithai Rabinowitch, Jihong Bai.

**Formal analysis:** Lisa Voelker, Bishal Upadhyaya.

**Funding acquisition:** Lisa Voelker, Denise M. Ferkey, Sarah Woldemariam, Noelle D. L'Etoile, Jihong Bai.

**Investigation:** Lisa Voelker, Bishal Upadhyaya.

**Methodology:** Lisa Voelker, Sarah Woldemariam.

**Resources:** Denise M. Ferkey, Sarah Woldemariam, Noelle D. L'Etoile, Ithai Rabinowitch.

**Supervision:** Jihong Bai.

**Visualization:** Lisa Voelker, Bishal Upadhyaya.

**Writing – original draft:** Lisa Voelker, Ithai Rabinowitch, Jihong Bai.

**Writing – review & editing:** Lisa Voelker, Bishal Upadhyaya, Denise M. Ferkey, Sarah Woldemariam, Noelle D. L'Etoile, Ithai Rabinowitch, Jihong Bai.

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
