## [Decision Letter · Decision Letter 0]

3 Sep 2019

Dear Dr Bai,

Thank you very much for submitting your Research Article entitled 'INX-18 and INX-19 play distinct roles in electrical synapses that modulate aversive behavior in Caenorhabditis elegans' to PLOS Genetics. Your manuscript was fully evaluated at the editorial level and by independent peer reviewers. The reviewers appreciated the attention to an important topic but identified some aspects of the manuscript that should be improved.

We therefore ask you to modify the manuscript according to the review recommendations before we can consider your manuscript for acceptance. Your revisions should address the specific points made by each reviewer.

[LINK]

Yours sincerely,

Coleen T. Murphy

Associate Editor

PLOS Genetics

Gregory P. Copenhaver

Editor-in-Chief

PLOS Genetics

Reviewer's Responses to Questions

**Comments to the Authors:**

Reviewer #1: In this manuscript, Lisa et al investigate the physiological function of electrical synapses in behavioral modulation in C. elegans. The authors have combined genetic manipulation, morphological analysis, behavioral test and imaging analysis to clearly show that the electrical synapses, including both INX-18 and INX-19, between two sensory neurons (ASH and ASK) modulate the quinine induced aversive response in ASH by controlling cGMP redistribution. This article provide a novel insight into the molecular basis for the the bias communication of intracellular signal cGMP between sensory neurons. Also, the article show that electrical synapses with different components have distinct roles in controlling cGMP levels in ASK and ASH. All experiments are well designed, and result provide strong support for their conclusion, so I am happy to support the paper published on Plos Genetics with minor changes.

Minor Comments:

1: The result shows that electrical synapses composed by inx-19 is permeable for cGMP, but not Ca2+. Is it possible the molecular different between inx-18 and inx-19 contribute to the permeability difference? Please comment.

2: The figure legends have lots of redundant information, especially for statistical analysis. It is better to put them in the method.

3: Unless I missed, I did not find which transgenic animals were used for fig 3.

4: The ASH promoter (osm-10) used in the paper also weakly expresses in ASI neuron, so it is possible that fluorescence disturb from ASI mess up the imaging single from ASH, especially for INX-19 puncta calculation in fig 3. Although I believe that the strong INX-19 fluorescence signalling comes from ASH based on my experience, it will be much clearer if the author clarify it in the manuscript.

5: In Fig 6 A and B, the marker of Y-axis, the change in fluorescence, is not clear. I believe it should be the percentage change based on GCaMP intensity calculation if I am correct. So please clarify it.

Reviewer #2: This manuscript from the Bai group addresses the role of electrical synapses in avoidance behavior. The manuscript investigates two genes inx-19 and inx-19, which are expressed in two different sensory neurons ASH and ASK.

The authors postulate that expression of the two proteins in necessary in the different sensory neurons for the communication to occur.

This is an excellent manuscript and I recommend the publication of the manuscript without any changes!

the experiments are well conducted with the appropriate controls.

I do have a couple of queries that I hope the authors can address:

1. the inx-19 allele (ky634) seems like a pleiotropic allele. IS the sensory responses to quinine in this allele due to its movement defects. Have the authors tested any other deterrents and do they observe similar locomotory defects.

2. Does the inx-19 puncta also show differences in the two different alleles ky634 and tm1896? This is not a suggestion of a new experiment but just a curiosity to know if different alleles elicit differences in expression.

3. Have the authors tested different concentrations of the quinine on the double mutant inx-19;inx-18?

I want to congratulate the authors of a great well written story and manuscript!

**Have all data underlying the figures and results presented in the manuscript been provided?**

Reviewer #1: None

Reviewer #2: Yes

PLOS authors have the option to publish the peer review history of their article (what does this mean?). If published, this will include your full peer review and any attached files.

Reviewer #1: Yes: Jie Liu

Reviewer #2: No

---

## [Decision Letter · Decision Letter 1]

4 Oct 2019

Dear Dr Bai,

We are pleased to inform you that your manuscript entitled "INX-18 and INX-19 play distinct roles in electrical synapses that modulate aversive behavior in Caenorhabditis elegans" has been editorially accepted for publication in PLOS Genetics. Congratulations!

Yours sincerely,

Coleen T. Murphy

Associate Editor

PLOS Genetics

Gregory P. Copenhaver

Editor-in-Chief

PLOS Genetics

Comments from the reviewers (if applicable):

Reviewer's Responses to Questions

**Comments to the Authors:**

Reviewer #1: The authors have addressed my concerns with the previous submission. I recommend an accept without any further changes.

**Have all data underlying the figures and results presented in the manuscript been provided?**

Reviewer #1: None

PLOS authors have the option to publish the peer review history of their article (what does this mean?). If published, this will include your full peer review and any attached files.

Reviewer #1: No

**Data Deposition**

http://datadryad.org/submit?journalID=pgenetics&manu=PGENETICS-D-19-01262R1

**Press Queries**

---

## [Editor Report · Acceptance letter]

24 Oct 2019

PGENETICS-D-19-01262R1 

INX-18 and INX-19 play distinct roles in electrical synapses that modulate aversive behavior in *Caenorhabditis elegans*

Dear Dr Bai, 

We are pleased to inform you that your manuscript entitled "INX-18 and INX-19 play distinct roles in electrical synapses that modulate aversive behavior in *Caenorhabditis elegans*" has been formally accepted for publication in PLOS Genetics! Your manuscript is now with our production department and you will be notified of the publication date in due course.

With kind regards,

Matt Lyles

PLOS Genetics

On behalf of:
